# Barriers to Using ESG Data for Investment Decisions

Bjorg Jonsdottir [1], Throstur Olaf Sigurjonsson [1,2,*], Lara Johannsdottir [1] and Stefan Wendt [3]

1   Faculty of Business Administration, University of Iceland, Saemundargata 2, 102 Reykjavik, Iceland; bjj25@hi.is (B.J.); laraj@hi.is (L.J.)
2   Department of Accounting, Copenhagen Business School, Solbjerg Plads, 2000 Frederiksberg, Denmark
3   Department of Business, Bifrost University, 311 Borgarnes, Iceland; stefanwendt@bifrost.is
*   Correspondence: olaf@hi.is

**Abstract:** Institutional investors who commit to integrating environmental, social and governance (ESG) aspects into investment decisions require ESG data of sufficient quality. However, concerns have risen over a lack of quality in ESG data, as outlined by the Global Reporting Initiative. The lack of quality in ESG data deters institutional investors from using the data for investment decisions. This study outlines the ESG data reporting process and explores where in the process quality concerns emerge. Semi-structured interviews are applied with professionals involved in ESG data analysis and reporting of listed companies, a rating agency and institutional investors. The results show that current barriers to using ESG data include a lack of materiality, accuracy and reliability. Interviewees agree that access to data collected by governmental institutions is lacking, and that companies' purchase of carbon credits raise questions about the reliability of ESG data. Companies hold contrasting views to the institutional investors on the useability of the data they disclose. The results enhance our understanding of the common and contrasting concerns about the lack of quality in ESG data. The results can be used as guide for companies, investors and regulators for actions to mitigate barriers related to the lack of quality in ESG reporting.

**Keywords:** ESG; data quality; reporting; investment decisions; integration barrier

## 1. Introduction

The United Nations (UN) launched the Six Principles for Responsible Investment (PRI) in April 2006 [1]. Corresponding to the first principle, institutional investors commit to integrating environmental, social and governance (ESG) aspects "into investment analysis and decision-making" [2]. The UNPRI includes guidance for the utilisation of ESG data, which identifies three approaches to integrating ESG data with financial considerations into investment analysis and decisions [1]. According to the first approach, screening, institutional investors filter investment options, ruling which ones can be included or need to be excluded according to the investor' standards or beliefs. Following the second approach, thematic investment, ESG data form a framework of an investment portfolio. According to the third approach, ESG integration, institutional investors fully integrate ESG into their investment selection models combined with all other material aspects. Institutional investors would thus adjust forecasted financials for the expected impact of ESG integration [1].

The integration of ESG aspects into investment decisions "starts with data" [3] (p. 2). ESG investments have increasingly attracted institutional investors because of the expected financial performance and corresponding capital inflows in, e.g., mutual funds that focus on ESG investments. Therefore, numerous institutional investors focus on ESG investments either because of specific clauses in their investment policy or due to regulatory requirements or pressure by stakeholders [4,5]. Institutional investors rely on the ESG data directly from companies "to hold them accountable for their ESG performance" [3] (p. 50), [6] and from sustainability rating agencies (SRAs) when building ESG investment portfolios [7,8].

Quality in ESG data is therefore paramount for making investment decisions, regardless of whether the data are self-reported by the companies, aggregated through SRAs or compiled through the latest information technologies, including artificial intelligence or machine learning tools [9,10]. The Global Reporting Initiative (GRI) [11] outlined ten principles that ESG data must contain to achieve "high quality sustainability reporting" (p. 7). The principles include four content principles, i.e., stakeholder inclusiveness, sustainability context, materiality and completeness, and six quality principles, i.e., accuracy, balance, clarity, comparability, reliability and timeliness.

Although the GRI principles were issued to improve the quality of ESG reporting [11], Kotsantonis et al. [6] questioned studies that have found "signals and meaningful relationships with economic outcomes given the inferior quality of the data" (p. 50). The quality issues extended into the ESG performance evaluation process [6,12,13]. Institutional investors still have concerns about quality issues in ESG data, which are a barrier to the use of the data for investment decision making [10,14].

According to Amel-Zadeh et al. [14] "little is known about how investors use ESG information" (p. 93). The literature on using ESG data for investment decisions has traditionally focused on the economic impact of the companies' ESG performance data [10,14–16]. Recent studies show, however, that present issues in ESG data include (1) a lack of relevance and materiality [14,15]; (2) a lack of accuracy for evaluating companies' actual sustainability performance, including "integrity, correctness; completeness and methodological consistency of the information reported" [9] (p. 495) [17]; (3) a lack of reliability in the self-assessment of ESG performance by companies, because of the increasing risk of whitewashing [18,19]; and (4) a lack of comparability, because the standards for ESG performance evaluations are misaligned [10].

Despite the existing literature on current quality-related concerns in ESG data, there is limited knowledge on where in the reporting process the quality-related concerns emerge and how the issues are revealed in the reporting process and impact the utility of ESG data for investment decision making.

The purpose of this study is to fill the gap in the knowledge by exploring the currently prevailing concerns about the lack of quality in ESG data through an ESG reporting process. The process initiates at the point of data collection and disclosure by companies, to ESG performance evaluations by (SRAs) and utilisation by institutional investors, as shown in Figure 1.

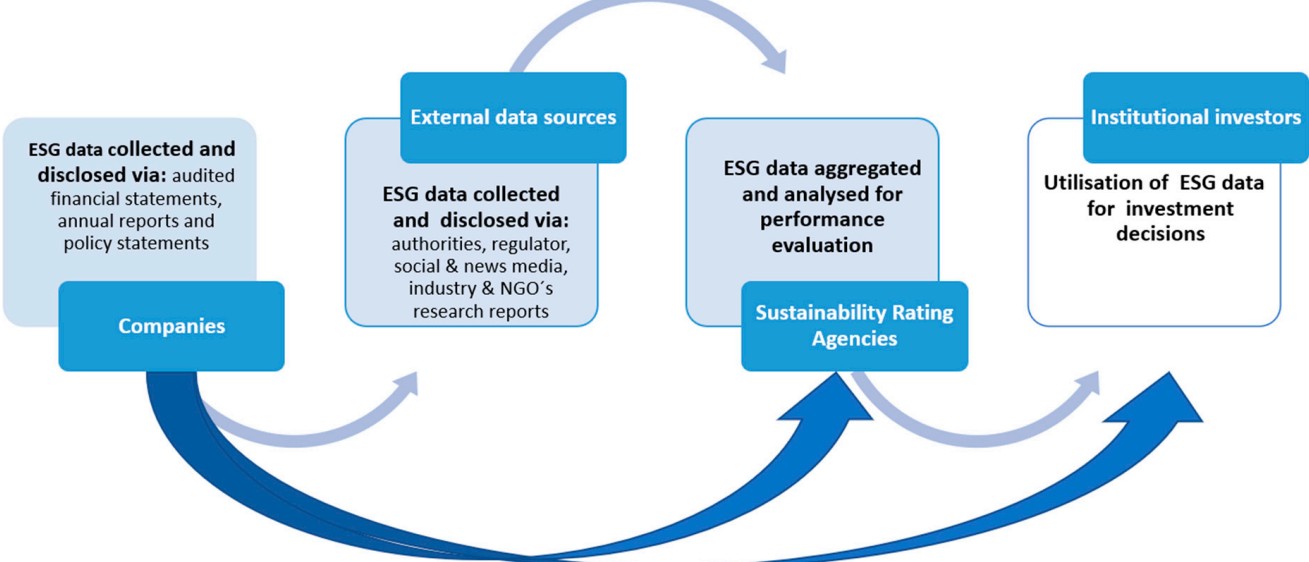

**Figure 1.** ESG reporting process. Source: Own illustration.

Approaching data collection from a reporting process perspective sheds light on where in the process quality concerns emerge. The study accordingly explores the following research question amongst companies, rating agency and institutional investors: Which quality-related barriers occurring in the ESG reporting process hinder the utilisation of ESG data for decision making for institutional investors?

The paper is organised as follows. Section 2 outlines an overview of the relevant literature. Section 3 describes the research method, followed by a presentation of main findings. The findings are deliberated in Section 5, followed by a concluding section which outlines the theoretical and practical implications of the study.

## 2. Literature Review

As of March 2022, there were over 4800 signatories to the PRI, with assets under management of more than USD 120 trillion in 2021 [20]. To integrate the ESG aspects into investment decisions, institutional investors require substantial amounts of quality ESG data to evaluate companies' performance in relation to the ESG aspects [8,21]. Consequently, there has been substantial growth in the number of companies that report on their sustainability performance by collecting and disclosing data on (1) the environmental aspect, such as carbon emissions, biodiversity, water consumption, waste management and protection of natural resources; (2) the social aspect, such as employee working conditions, diversity and equal pay; and (3) the governance aspect, which includes diversity of the companies' leadership, anticorruption programs, executive pay, shareholder rights, audits and internal controls [14,22].

In addition to the principles for sustainability report quality and content issued by the GRI [11] outlined in the previous section, companies can use various frameworks and standards for ESG reporting. The most recognised ones include the International Integrated Reporting Council (IIRC) framework, the Global Reporting Initiative (GRI), Principles of Responsible Banking (PRB), Principles of Responsible Investing (PRI), the Climate-related Financial Disclosures (TCFD), the Sustainability Accounting Standards Board (SASB) framework and the UN Global Compact [16,23,24]. GRI is the most widely adopted framework for ESG reporting [24].

SRAs include specialist rating agencies, investment banks' analysis departments, operators of securities indices, credit rating agencies as well as non-governmental organisations [25–27]. The sustainability ratings are employed to benchmark the companies against their peers. SRAs are therefore instrumental in defining the features of ESG portfolios in terms of risk exposure and how the monetary impact of the ESG aspects is assessed [28]. SRAs require accurate, transparent and reliable ESG data to evaluate companies' sustainability performance [17]. SRAs aggregate the ESG data provided by the companies and from external sources to evaluate how the reporting companies are performing in relation to the ESG aspects, as shown in Figure 1. The publicly available data sources used for collecting ESG-related data include audited financial statements and the companies' annual reports and policy statements. In addition, social and news media, companies' websites, industry research reports and other publicly accessible data sources are also used for gathering and aggregating ESG data [3,13]. Many of the leading SRAs give companies an opportunity to provide feedback or additional data to their reports [29].

Despite the rising popularity in ESG investing followed by a vast increase in the volume and variety of available ESG data, a substantial proportion of the investment community is not pursuing ESG aspects as part of their investment strategy [30]. Several studies have suggested that the vast variety and volume of available ESG data has resulted in poor data quality [3,6,14], because it becomes "increasingly challenging for investors to efficiently determine the quality of those datasets in any principled way" [17] (p. 7). Although the purpose of GRI principles is to improve the quality of ESG data, it is still a voluntary requirement; companies can choose which reporting framework to use for collecting and disclosing their ESG data [31–33].

The Corporate Reporting Dialogue [33] is an initiative organised by the Integrated Reporting Framework for the purpose of achieving "greater coherence, consistency and comparability between corporate reporting frameworks, standards and related requirements" [33] (p. 2). The Corporate Reporting Dialogue states that:

"There is a large and stultifying ecosystem of regulation, voluntary frameworks and standards, and surveys and questionnaires seeking the disclosure of ESG information. It is clear from the stakeholder engagement that report preparers lack the resources to effectively navigate and respond to these reporting requests, leaving report users with disclosures that are not as comprehensive, consistent, or comparable as desired" (p. 28).

The term "quality" in relation to ESG data is multifaceted, and it is complex to shed a light on which of the various quality aspects, as defined by GRI, are currently relevant and hindering the utilisation of the ESG data [10]. Currently, institutional investors are mainly concerned about ESG data lacking quality aspects related to data materiality, accuracy, reliability and comparability [16].

### 2.1. Materiality

The GRI principles on quality define material data as relevant on matters "that can reasonably be considered important for reflecting the organization's economic, environmental, and social impacts, or influencing the decisions of stakeholders" (p. 10). Despite the GRI principle, the challenge to obtain high-quality data on material ESG exposures persists. Studies by Khan et al. [15] and Amel-Zadeh et al. [14] showed that although companies have identified the strategic importance of ESG reporting and increased the volume of the ESG data they disclosed, the data are commonly over-generalised and thus irrelevant and immaterial for making investment decisions. According to Danisch [34], the significance of materiality in ESG data might even render the GRI quality requirement for completeness in ESG data less important if the content disclosed is focused on materiality analysis. Another reason for over-generalisation and lack of relevance in ESG data is a "tick the box" approach by companies which emerges because of the exponential rise in the number of SRAs and the corresponding increase in demand for ESG data. This shortcut is taken due to a lack of resources or motivation to respond to the increasing data demand [6,13]. A study by Kotsantonis et al. [6] showed that only half of the fifty listed Fortune 500 companies included in their study reported on having a health and safety policy, and that only about 15% disclose their lost time incident rates and workplace fatalities, which is significant because "employee health and safety is a material ESG issue for 9 out of the 11 sectors, according to SASB's framework" (p. 50). Moreover, Kotsantonis et al. [6] claimed that there is "currently no agreed method on how to handle diversified businesses, in terms of which ESG issues are material to them" (p. 54).

### 2.2. Accuracy

According to the GRI principle on accuracy, ESG data disclosed by companies should be "sufficiently accurate and detailed for stakeholders to assess the reporting organization's performance". Kotsantonis et al. [6] showed that because of the lack of alignment in reporting frameworks, the reporting process becomes overly complex and resource heavy. Consequently, because of the complexity and heavy resource demand, the ESG data disclosed by companies become inaccurate as the companies might avoid ESG data disclosure, and those that do disclose might provide inferior quality ESG data, leaving large gaps in the data for key ESG aspects [6].

Adding to the lack of accuracy in ESG reporting is the "difficulty in accurately understanding the [overwhelming GRI] requirements and preparing a complete report" (p. 285), which causes reporters' fatigue amongst the reporting companies' employees, as revealed in a study by Dissanayake et al. [35]. Furthermore, In et al. [10] added that because of the resource-heavy process required to aggregate ESG data and "verify the accuracy" (p. 246), institutional investors refrain from gathering more data to enhance data accuracy.

The study by Kotsantonis et al. [6] also showed that the companies used over twenty different terminologies and metrics for reporting on the same ESG aspect. The inaccuracy of the disclosed data leads to inconsistencies and therefore to a lack of comparability among companies' ESG performance evaluations [14,30].

Del Giudice et al. [36] claimed that ESG data lacked accuracy because "non-financial reporting is still in its infancy" (p. 1) and the reporting thus lacks. They added that "[t]his exposes the ESG information released by companies to varying of accuracy, which can ultimately impact the reliability of ESG scores" (p. 1).

### 2.3. Reliability

The GRI reliability principle requires that companies "gather, record, compile, analyze, and report information and processes used in the preparation of the report in a way that they can be subject to examination, and that establishes the quality and materiality of the information" (p. 15). Nevertheless, institutional investors consider self-assessment of ESG performance unreliable because companies are prone to report only the positive aspects of their activities. ESG data will therefore not correspond to their actual behaviour, which increases the risk of window dressing [13,18,19,37].

Companies that perform poorly often attempt to whitewash the ESG data using language that is imprecise but optimistic [19,38]. Similarly, companies with riskier operations tend to augment their ESG data disclosure as window dressing, but this also enhances the transparency of their operations and decreases the companies' risk [39]. Cho et al. [18] define the concept of window dressing as the misleading use of ESG data disclosure, designed to create a favourable view without the intention to honour the disclosure by implementing actions.

To mitigate the risk of whitewashing, many of the global SRAs, including MSCI, RepRisk and Refinitiv, commonly aggregate ESG data only from external, public data sources [3,13,29]. The SRAs collect external data via sources including audited financial statements and annual reports of the disclosing companies, media information, companies' websites or other publicly accessible data sources [3,13,39].

Moreover, because of the distrust in the legitimacy of the self-assessed performance related to the ESG aspects, institutional investors are calling for mandatory ESG data disclosure [12,40,41] and auditing of ESG reports, based on global standards by external assurance providers, such as accounting or consulting firms [22]. The shift from voluntary to mandatory sustainability reporting emerged via the European Union (EU) Directive 95/2014, which enforces large companies to disclose sustainability data [31,42]. Although empirical evidence shows that mandatory ESG data disclosure is believed to improve the quality in ESG reporting [43,44], evidence also shows that ESG data still lack reliability because large companies lack robust internal controls to ensure reliable ESG disclosure [45], that gaps still occur in the ESG datasets and, importantly, that the risk of whitewashing still prevails [46]. In addition, according to La Torre et al. [40], mandatory reporting increases a facile approach to ESG data disclosure by companies and thus escalates the "inconsistencies between corporate talk and action" (p. 9). Moreover, the lack of aligned standards in ESG performance evaluations interferes with making the evaluations comparable if the disclosure is still voluntary in other parts of the world [47].

### 2.4. Comparability

Comparability, according to the GRI, refers to the consistent collection and disclosure of ESG data by companies. The data "shall be presented in a manner that enables stakeholders to analyze changes in the organization's performance over time, and that could support analysis relative to other organizations" (p. 14). In a survey of 652 investment professionals [14], the most significant challenge facing investors in utilising ESG data as a base for investment decisions related to a lack of cross-company comparability. The lack of comparability was a consequence of the multidimensionality in the standards governing the ESG reporting, which makes it difficult for institutional investors to identify which

outcomes were related to ESG performance [12,48]. In et al. [10] also stated that "today's ESG ratings are inconsistent and incomparable, as they are not relying on shared theoretical foundations, nor sharing the common reporting standards" (p. 240).

Furthermore, the heterogenous missions and goals that drive institutional investors to seek ESG data, as well as sector-specific approaches and national preferences, have made it difficult to define acknowledged standards and frameworks for constructing a comparable and dependable ESG investable universe [7,49].

## 3. Research Method

The semi-structured interview method was chosen to enable the interviewer to control the conversation [50,51]. Nine interviewees were purposefully chosen to ensure their involvement in the implementation of ESG reporting and utilisation in their organisations. The interviewees were therefore likely to possess the knowledge sought for the purpose of the study [52–54]. The interviewees were from companies in different industries listed on the Nasdaq Nordic-Iceland stock exchange, a rating agency, and institutional investors to reflect key actors in the ESG reporting process and to allow us to follow ESG data through the process, as shown in Table 1.

**Table 1.** Overview of interviewees.

| Organisation Type: | Role: |
|---|---|
| Production Company | Sustainability Specialist |
| Transport Company | Sustainability Specialist |
| Transport Company | Manager |
| Commercial bank | Sustainability Specialist |
| Commercial bank | Sustainability Specialist |
| Sustainability rating agency | Manager |
| Pension fund | Fund Manager |
| Pension fund | Senior Administrative Manager |
| Pension fund | Fund Manager |
| Pension fund | CFO |

As suggested by Creswell et al. [52] and Patton [53], the interviewees were selected from a list of high-level sustainability managers and experts within the entities. The commercial banks reported sustainability data from their operations, but they are also institutional investors. Therefore, the participating commercial banks were able provide data from the perspectives of the ESG data collection and disclosure procedures as well as the utilisation of ESG data for investment decisions.

The total interview time was 7.32 h for nine interviews. An interview guide was followed during the interviews to achieve optimal use of interview time and to keep the interviewees focused on the topic [54]. After each interview, the interview guide was revisited to evaluate the need for any adjustment. A qualitative method was used for the study to describe and explain what the interviewees revealed and to avoid generalising the results. In this sense, repetition of the study would not necessarily give the same results. However, this does not detract from the results of the study. To ensure reliability and consistency in the data, an audit trail was used to describe exactly how the data collection was conducted [55].

Interviews were transcribed and analysed thematically using the atlas.ti software to identify the main themes arising from the interviews. Each sentence was analysed and coded. Over 500 codes were obtained and subsequently sorted into categories and labelled to create a descriptive concept for each category. The main categories that emerged from the from the interview data [56–59] were data accuracy, reliability, comparability and data materiality. The results were drawn from the interview data and are presented in the following section.

## 4. Results

The institutional investors who participated in the study were all signatories to the PRI and were integrating ESG data into their investment analysis and decision making. The pension funds had also incorporated the governance aspect into their policies and practices but not the environmental and social aspects. Accordingly, interviewees actively engaged as shareholders with the board and managers of the invested companies. The shareholder engagement occurred mostly during annual meetings but also, on occasion, directly with company managers. Some interviewees claimed that Iceland is two to three years behind the rest of Europe in the integration of ESG aspects into investment policy, a key reason being the deep impact of the 2008 banking crisis. As a result, their focus to date was on the governance aspect.

The commercial banks were further advanced in using ESG data for investment decisions than the pension funds and were building a framework to evaluate the sustainability of the asset and lending portfolios. Interviewees from the pension funds, however, had not formally engaged in issues related to the ESG aspects beyond the governance aspect, although they invested a portion of their funds in foreign sustainability equity funds. The pension fund interviewees did so by choosing which ESG strategy corresponded with their investment strategy and/or which ESG performance measurement methodology most appealed to them.

One of the pension funds was further ahead than the other pension funds in the utilisation of ESG data for foreign investments, because they cooperated with a global SRA. For domestic investments, however, the pension fund managers looked at ESG data on a case-by-case basis. Most of the pension funds were thus still in the initial stages using ESG data; these interviewees were contemplating which of the ESG integration approaches to take and whether and how the integration of ESG aspects into investment decisions would impact their main duty, which was to deliver returns to their shareholders. ESG data thus served as a secondary perspective on domestic investments and were not critical for a decision to invest or not.

All the participating companies used Nasdaq guidelines as a framework for sustainability reporting; three of the interviewees used either GRI or TFCD guidelines as well. One company interviewee claimed to extract ESG data from the GRI report into the Nasdaq report to produce "fast boiled" information for investors who do not have time to go through the GRI report. Another company interviewee claimed to use the TFCD guidelines because they were more comprehensive than the Nasdaq guidelines and were better suited for providing a forward-looking perspective on ESG data. The company interviewee claimed that the climate crisis was a financial risk for institutional investors and, as such, relevant to the survival of the companies they invest in. Two company interviewees claimed that the Nasdaq reporting guidelines were better suited for companies that were at the beginning of the ESG reporting integration process, rather than organisations that were further ahead in the process.

The SRA interviewee, however, explained that the reason for the wide application of the Nasdaq guidelines was because when ESG reporting first emerged, companies were overwhelmed by the enormity of data required. Moreover, the main challenge encountered by SRAs was the lack of standardisation in the field, particularly between the methodologies applied by the various SRAs.

Asked if the ESG data currently disclosed by the companies were useable for institutional investors for investment decisions, the company interviewees answered that the data should be usable for institutional investors because the data are collected and disclosed per GRI and Nasdaq ESG guidelines. The investors "should [therefore] be able to compare us with others", claimed one company interviewee.

### 4.1. Lack of Materiality

In an interview on ESG data materiality, a pension fund interviewee described ESG data disclosure from an investor's point of view:

> "Not focused enough on what is important in the activities of the companies in question [and that disclosing companies sometimes disclosed] all kinds of data to be able to check the box, the data does not necessarily have value or is catalytic."

Fund managers watched the numbers and could see how they evolve from one year to the next, but that information was not material for making an investment decision. The commercial bank interviewees upheld the notion of irrelevance and claimed that companies reported ESG data generated within their own operations and that data were often irrelevant to investors. Internal data, claimed the commercial bank interviewees, are just a "drop in the ocean", whereas their main environmental and social impact resides in the asset portfolios and external operations, e.g., the supply chain.

According to a pension fund interviewee, the relevant information should cover environmental and social impact and actions related to the core business of the companies, which differs between industries. However, this type of data is not easily accessible because:

> "When investing in an insurance company or a bank, reading about the water consumption at the company in question has little relevance. It would be different if it was, say, a soft drink manufacturer or a fish processing factory."

The findings also suggest that companies are under pressure from SRAs that demand further ESG data, which pushes companies towards a "tick-the-box" approach to ESG reporting. A company interviewee described constantly "playing defence" against the SRAs, because they give the companies limited time to give feedback on sustainability evaluation reports. In addition, there is constant pressure to join various ESG data platforms; the SRAs claim that they have an ESG data portal with:

> "Hundreds of investors who subscribe to the data, and it only costs a million to join ... this is a big money business. ... it is just a little bit oppressive sometimes."

### 4.2. Lack of Accuracy

A company interviewee outlined how the challenges in disclosing ESG data were related to

> "Setting the framework, understanding where the data is coming from, consistency in the terminology of the variables. However, maintaining consistent, accurate and comparable data year on year is a huge challenge."

Other interviewees in the study described the challenges related to "mixed motives" in investment options. One pension fund interviewee explained the "mixed motives" inherent in a case of discarded plastics. Iceland currently exports its plastic waste to Sweden. This poses the question if it would make sense to set up a plastic recycling factory in Iceland:

> "Would anyone want to invest in such a factory? And on what perspectives are you going to base that investment? This is a polluting factory."

Similarly, a commercial bank interviewee claimed to struggle with the meaning inherent in sustainability certifications and mentioned as an example the Icelandic fishing industry. The Marine Stewardship Council (MSC) certifies if the Icelandic fishing fleet is operating in a sustainable way. Correspondingly, the fishing industry also reports on fuel consumption in their operations, which the companies claim to have reduced in the past decade. Although obtaining the certification is a positive step towards sustainability, the commercial bank interviewee said that:

> "The entire fishing industry is however not an environmentally friendly business. The whole fleet is run on fossil fuels."

The commercial bank interviewee also described how complex it was to delve into the reports for a better understanding and to:

> "Analyse the numbers to distinguish, for example, whether a fishing company that gets a loan is using that loan to buy fossil fuels . . . then you often end up hitting walls . . . It is difficult to determine if a fishing company that receives a loan uses the money to invest in more environmentally safe energy supplies."

Institutional investors and the reporting companies also require access to ESG data from sources outside the companies. A commercial bank and two pension fund interviewees stated that they faced challenges when considering investments in, e.g., eco-friendly housing (built to reduce the carbon footprint and use energy sources efficiently), or a more sustainable fishing industry. Governmental institutions, including the Icelandic Directorate of Fisheries, the National Energy Authority, the Environmental Agency and the Housing and Construction Authority, collected data from various businesses to seek to enhance sustainability reporting by companies. The Icelandic authorities could play a bigger role in leading a scaled-up utilisation of ESG data for reporting and utilisation.

A commercial bank interviewee described how the commercial bank had met with various governmental authorities to see whether ESG data, especially on climate resilience, existed.

The authorities could, as mentioned by a pension fund interviewee, help improve the accuracy and reliability of ESG data by making the data that are aggregated by the governmental institutions more accessible for ESG reporting actors, and thus enhance the data quality disclosed by the companies and potentially reduce the costs incurred in data collection.

### 4.3. Lack of Reliability

The institutional investors and one company interviewee stated that whitewashing, also referred to as greenwashing and window dressing by the interviewees, was prevalent in ESG reporting. The interviewees claimed that whitewashing was a paramount contributor to the institutional investors' perception that ESG data are unreliable and thus a key barrier to impactful utilisation. One pension fund interviewee stated that "whilst sustainability reports from every single company are very good, that may not be the most solid foundation" from the investors' point of view.

Another pension fund interviewee claimed that since the performance evaluation of companies is entirely based on ESG data disclosed by the companies themselves, and not from external sources, the inherent risk of whitewashing is high. This is " . . . naturally very dangerous." Therefore, the pension fund chose to cooperate with one of the global SRAs, because it aggregated ESG data mostly from sources outside the companies and not directly from the companies themselves.

In addition, a pension fund interviewee stated that the SRA they co-operated with has:

> "Access to a lot of data from the tax authorities, for example, or from some governmental agencies who have at least 25 years of data published by the boards and shareholders' meeting data and therefore we find it reassuring how the data is collected."

The third pension fund manager described how:

> "An industry is emerging with the explosion in the trade in ESG data, within which the risk of window-dressing is high; therefore, the fund's task is to sort the wheat from the chaff."

The interviewees stated that data auditing and assurance by external data assurers mitigated the risk of "whitewashing", but one company interviewee claimed that:

> "ESG data is not audited because it is non-financial information. Although the ESG auditing industry in Iceland is emerging, it is currently limited to environmental data, there is still no true accountability, and the risk of greenwashing exists."

While one of the participating commercial banks had purchased certified carbon credits to offset the operations and became certified as "carbon neutral", a company interviewee claimed that whitewashing is inherent in the transactions with carbon credits and therefore questionable in terms of the goals for zero carbon emissions. The company interviewee claimed that:

> "There is a common misunderstanding in the market regarding carbon. At the end of the day, companies are not becoming greener if they purchase carbon credits for the purpose of claiming they are carbon neutral."

ESG data provide a status report on company performance in terms of sustainability. If a manufacturing company purchases carbon credits, as does the end user of the component, it will be carbon neutralised twice. The component and the end products, however, will not become "greener" at the end of the day, as claimed by a company interviewee. This defeats the objective of becoming carbon neutral and, consequently:

> "There is a big issue about 'green washing' in Iceland because companies misunderstand what being carbon neutral means."

Two of the pension fund interviewees seconded the companies' statements regarding carbon credits and talked about how the market transactions with carbon credits has become a money-making industry, thus reducing the pension funds' trust in sustainability data.

Further on the topic of ESG data reliability, one pension fund interviewee stated that the "Holy Grail" is to ensure that there are benefits for the companies and investors. For instance, when companies issue green bonds, investors can offer specific loan agreement terms with lower interest rates that are linked to achieving specific ESG performance goals. The companies have thus "linked sustainability development to financial benefits", which requires the companies in question to disclose reliable data on their ESG performance.

*4.4. Lack of Comparability*

One pension fund interviewee described how the same companies sometimes received dissimilar ESG performance evaluations from three different SRAs, which is:

> " . . . thought-provoking when the same companies receive a poor evaluation from one SRA, moderate from another and then a superior performance evaluation from the third SRA. What are you going to do here?"

The pension fund interviewee also stated that because of the dissimilarities in the performance evaluations, the pension fund has:

> " . . . not yet started here in Iceland to be able to somehow trust or properly use any third party as some kind of data analyst, neither for the listed nor the unlisted companies."

Despite the dissimilarities and, as a result, the lack of comparability in the companies' performance evaluations, the pension funds confront these concerns by studying the methodology behind the ESG reports and ratings and choose either the one that best corresponds to their investment strategy or the one they most agree with. Another pension fund interviewee added that fund managers look at various numbers that are not completely "clear-cut"; the expectations relating to the ESG data and ratings are no different in this respect. The representatives of the Icelandic pension funds included in the study are therefore not as concerned with the lack of comparability in ESG data as are their foreign counterparts.

## 5. Discussion

The key findings in this study show that the paramount quality-related concerns in ESG data relate to the lack of materiality, accuracy and reliability of the data collected and disclosed by companies for institutional investors. The concerns arise because of misalignment in the frameworks available for companies to follow when collecting and disclosing ESG data [5,35], amongst other reasons.

The companies' interviewees claim to be overly burdened with "ticking boxes" for the annual reports and fending off SRAs. Consequently, the companies leave large gaps in the disclosed data or disclose data that are immaterial from an institutional investors point of view. These issues are in line with the literature [6,13,14,34].

Corresponding to Cort et al. [9], the lack of accuracy in ESG data is also a barrier for the utilisation of ESG data because the institutional investors, as stated by a commercial bank interviewee, are unable to testify to the significance of companies' sustainability certificates, e.g., certified fishing (MSC) for the fishing industry. This issue of lack of accuracy has, so far, not been emphasised in the literature.

The participating institutional investors claim that they distrust the companies' self-assessment of their sustainability performance, which is a barrier to the utilisation of ESG data for investment decisions. The institutional investors claim that the ESG data do not reveal the companies' actual behaviour and the risk of window dressing is therefore prevalent [13,18,19]. Moreover, institutional investors' calls for external auditing [23] to mitigate the risk of whitewashing is in line with the literature.

Concerns about the lack of comparability in ESG performance evaluations was not as significant for most of the pension funds interviewed, as outlined in the literature [14,37], although the interviewees recognised that the issues existed. Most of the pension funds claimed to resolve the issues of incomparability in ESG performance evaluations by choosing the ESG investment policy or performance measurement methodology they most agreed with.

Other quality-related concerns revealed in this study, which do not align with the literature, include the carbon credits market. The carbon credit topic kept recurring amongst the companies and pension fund interviewees during the interviews, in relation to data on carbon offsetting. There was a consensus amongst some companies and pension fund interviewees that purchasing carbon credits increased the risk of whitewashing because, at the end of the day, the purchased credits neither offset the companies' carbon emissions nor result in greener end products.

Additional findings which were not found in the literature include the outlining by two pension fund interviewees of loan agreements that tie interest rates to specific sustainability goals. The loan agreement terms help mitigate the risk of whitewashing and create a financial incentive for companies to improve the quality in their reporting. The institutional investors also call for more extensive support from governmental authorities in terms of making the data the institutions (e.g., Icelandic Inland Revenues and Customs) collect from the companies more accessible and thus reducing the gaps in the ESG datasets.

Moreover, not in line with the literature was the apparent consensus between the companies and the pension funds interviewees interviewed when asked about the useability of the disclosed data for investors. As stated by the companies' interviewees, the ESG data disclosed are useable because they follow the GRI or Nasdaq reporting frameworks. The findings, however, highlight various quality-related concerns, such as a lack of materiality, accuracy and reliability, that bar institutional investors from using the ESG data for investment decisions.

## 6. Conclusions

This study analysed the quality in ESG data and identified which quality-related aspects, of the aspects defined in the ten GRI principles on quality, are currently lacking. Studies on the lacking quality in ESG data and how the quality issues appear among ESG actors in an ESG reporting process are still relatively scarce, although interest in the subject is growing apace. This study therefore adds to the knowledge on the significance of quality in ESG data and how the lack of quality related to materiality, accuracy and reliability emerges from the companies' data collection and disclosure. Lack of ESG data comparability is a concern amongst the participating institutional investors but not a hindrance to using the data for investment decisions because they analyse the methodology behind the data and choose to follow the one, they most agree with. Practical implications are that management

boards, fund managers and SRAs need to be aware that the transactions with carbon credits raise concerns in relation to the reliability of the companies' ESG data disclosures. They also need to partner with regulators and issuers of sustainability certificates to shed a light on the accurate meaning and impact of companies' sustainability certificates, e.g., MSC in the fishing industry. The companies and institutional investors need to consider the practical implications of a scaled-up utilisation of loan agreement terms that tie the interest rates to the achievement of specific sustainability goals. The companies, SRAs and investors need to partner up and call on governmental institutions for more extensive support in terms of making available the data the institutions collect from the companies.

The novelty of the study resides in the research approach, which follows the ESG reporting process to provide valuable insights into the common and controversial concerns related to the lack of quality in ESG data. The approach also explores where within the process the quality-related issues emerge. Moreover, the reporting process approach clarifies the links between the companies, SRAs and institutional investors within the reporting process. The study is, however, limited by the small number of interviewees especially related to SRAs, although the interviewees were key players in their fields and as such can provide valuable data.

Future studies on this topic should therefore explore how the quality-related concerns in ESG data disclosed by companies appear in ESG performance evaluations by SRAs and/or institutional investors. Moreover, it could be worth exploring further whether and, if so, how the market transactions with carbon credits negatively impact institutional investors' reliance on ESG data. Another topic to explore is whether the utilisation of loan agreement terms tied to ESG goals as a tool to mitigate the risk of whitewashing in ESG data could be scaled up. In addition, further understanding is needed on the wider significance, meaning and impact of sustainability certifications, as well as enhancing the role of governmental authorities in the ESG reporting process.

**Author Contributions:** Conceptualisation, B.J. and T.O.S.; methodology, B.J.; validation, T.O.S., L.J. and S.W.; writing—original draft preparation, B.J.; writing—review and editing, B.J., T.O.S., L.J. and S.W.; visualisation, B.J., T.O.S., L.J. and S.W.; supervision, T.O.S.; project administration, T.O.S. All authors have read and agreed to the published version of the manuscript.

**Funding:** This research received no external funding.

**Informed Consent Statement:** Informed consent was obtained from all subjects involved in the study.

**Data Availability Statement:** Anonymous and confidential interviews were conducted.

**Acknowledgments:** We acknowledge the interviewees for their valuable insights to the research topic.

**Conflicts of Interest:** The authors declare no conflict of interest.

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
