# Peer review of "Barriers to Using ESG Data for Investment Decisions"

_sustainability, doi:10.3390/su14095157_

Round 1

Reviewer 1 Report

1- The paper needs more justification in terms of the research gap and contributions. Several studies have been conducted investigating the barriers to sustainability issues from different aspects (see [1]). Accordingly, the authors need to justify their contribution to the strand literature, theory, and practice.

2- The results of the study are organised in 4 sub-sections. These sections are almost related to the traditional characteristics of accounting information. Any justification for presenting these specific heads. Is there any relationship between the presentation of these sections and the main focus of the interviews?

3- The authors have successfully articulated three research questions. However, the discussion of the results are not linked to these questions.

4- There is a need to focus on specific responses given by the respondents on each issue. For example, the following text is quested from [1]:

The issue of regulations for sustainability reporting is likely to contribute to gaps in the understanding of the concept of sustainability and sustainability reporting amongst the general public and particularly employees of the companies, as indicated by the following extract from the interviews: If you are the company, and if you only see in making the report is the expense side, you will not push through with it. So, another hindrance I see is that among those who are publishing a GRI certified (sic) report, no one is disclosing the benefits that they derive from publishing the report. I hope that more companies will be encouraged to produce a sustainability report, I hope there will be a forum where they can share with the public the benefits of producing a sustainability report (Interviewee P2). Consequently, there is a propensity for sustainability practices to be associated with values and traditions held in society and religious concepts in developing countries. Usually the Filipino asks before he jumps in, ‘What’s in it for me?’ If he sees he’s not going to get anything from it, so why would I spend time, resources, and stress people out doing it if at the end of the day I will not gain anything from it? So, this is one of my observations that is happening here in the Philippines” (Interviewee P2)

The authors need to be more specific to discuss each dimension of their research separately taking into consideration the respondents’ responses.

5- The authors claim that “ Interviews were coded open coding; each sentence was analysed each sentence and 267 each unit of data was labelled to create a descriptive concept for each unit. Simultane- 268 ously, a memorandum was created while coding, using the comments that were subse- 269 quently exported from the document. Overall, 505 codes were obtained from the inter- 270 views and axis coding was applied, to generate themes with respect to the notes [59-63]”, however, the results and discussion presented in the manuscript do not show any quantitative or descriptive information. The authors are suggested to report their coding, labelling, and descriptive.    

6- The authors need to explain the main focus of the barriers. This focus should link the research questions with the results of the study.

[1] Dissanayake, D., Kuruppu, S., Qian, W. and Tilt, C. (2021), "Barriers for sustainability reporting: evidence from Indo-Pacific region", Meditari Accountancy Research, Vol. 29 No. 2, pp. 264-293. https://doi.org/10.1108/MEDAR-01-2020-0703

Author Response

Dear reviewer, please find our responses in the attached word document. We highly appreciate your constrictive feedback. Best regards, the authors. 

Reviewer 2 Report

This study outlines an ESG data reporting process and explores where in the process the quality-related concerns emerge via semi-structured interviews with employees involved in  ESG data analysis and reporting within companies, rating agency and amongst institutional investors.  For exploring the barriers to using ESG data for investment decisions is an interesting case study. The authors have to clarify more  theoretical literature within that scope. Specifically, lack of accuracy part have to provide nore detailed information. The paper is readable. However, it may be helpful to gain assistance from an editor to refine sentence structuring across the paper.

Author Response

(The authors gave the same response as above.)

Round 2

Reviewer 1 Report

Accepted